# Extreme amyloid polymorphism in *Staphylococcus aureus* virulent PSMα peptides

Nir Salinas [1], Jacques-Philippe Colletier[2], Asher Moshe[1,3] & Meytal Landau [1]

Members of the *Staphylococcus aureus* phenol-soluble modulin (PSM) peptide family are secreted as functional amyloids that serve diverse roles in pathogenicity and may be present as full-length peptides or as naturally occurring truncations. We recently showed that the activity of PSMα3, the most toxic member, stems from the formation of cross-α fibrils, which are at variance with the cross-β fibrils linked with eukaryotic amyloid pathologies. Here, we show that PSMα1 and PSMα4, involved in biofilm structuring, form canonical cross-β amyloid fibrils wherein β-sheets tightly mate through steric zipper interfaces, conferring high stability. Contrastingly, a truncated PSMα3 has antibacterial activity, forms reversible fibrils, and reveals two polymorphic and atypical β-rich fibril architectures. These architectures are radically different from both the cross-α fibrils formed by full-length PSMα3, and from the canonical cross-β fibrils. Our results point to structural plasticity being at the basis of the functional diversity exhibited by *S. aureus* PSMαs.

[1] Department of Biology, Technion-Israel Institute of Technology, Haifa 3200003, Israel. [2] Institut de Biologie Structurale, Université Grenoble Alpes, Centre National de la Recherche Scientifique (CNRS), Commissariat à l'énergie atomique et aux énergies alternatives (CEA), Grenoble 38044, France. [3] Present address: School of Molecular Cell Biology & Biotechnology, George S. Wise Faculty of Life Sciences, Tel Aviv University, Tel-Aviv 6997801, Israel. Correspondence and requests for materials should be addressed to M.L.  (email: mlandau@technion.ac.il)

Amyloids designate peptides and proteins capable of self-assembling into structured oligomers and fibrils, and they are mostly known for their involvement in fatal neuro-degenerative diseases, such as Alzheimer's and Parkinson's diseases[1]. Some amyloids are functional, in that they participate in specific physiological activities. In humans, functional amyloids partake in immunity, reproduction and hormone secretion[2–4]. In microorganisms, they act as key virulence factors, and may thus represent novel targets for antibacterial drugs[5,6]. For example, amyloid fibrils are found in the self-produced polymeric matrix that embeds biofilm-forming bacteria, where they act as a physical barrier that increases their resilience and resistance to antimicrobial drugs[5–8] and to the immune system[9]. Functional bacterial amyloids may also act as toxins, killing non-self-cells and thereby increasing virulence[6,7,10]. The structural hallmarks of functional amyloids—if any—and how they can be distinguished from disease-associated amyloids remain unclear. To date, only a single atomic resolution structure of a functional bacterial amyloid has been made available, namely, that of phenol-soluble modulin α3 (PSMα3), the most cytotoxic member of the *Staphylococcus aureus* PSMs peptide family. The structure of the full-length PSMα3 revealed cross-α fibrils[10], a newly discovered mode of self-assembly characterized by the piling of α-helices perpendicular to the fibril axis, in place of β-strands in canonical cross-β amyloid fibrils. Cross-α fibrils[10] form through the tight mating of α-helical sheets, just as cross-β fibrils form through the tight mating of β-sheets[11]; it was thus proposed that they are amyloid-like.

*Staphylococcus aureus* is a prominent cause of threatening infections[12] and its PSM family members[13] serve as key virulence factors that stimulate inflammatory responses, alter the host cell cycle, lyse human cells and contribute to biofilm structuring[14,15]. High expression of PSMαs, which are four peptides of about 20-residues in length, increases virulence potential of methicillin-resistant *S. aureus* (MRSA)[16]. Amyloid aggregation plays roles in PSM activities in *S. aureus*. For example, fibrillation of PSMα1 promotes biofilm stability by preventing disassembly by matrix-degrading enzymes and mechanical stress[14]. In addition, fibrillation of PSMα3, enhances its toxicity against human cells[10]. Still, the extent of amyloid fibrillation in all PSM activities remains unclear, especially since PSMs are known to undergo truncation in vivo in response to various external stimuli, yielding truncated PSMs with new functions such as antibacterial activities[17–21]. As each of the truncated PSMs may adopt a different type of amyloid architecture, the array of structural species produced by PSMs is much larger than the actual number of PSMs. In the present study, we investigated the structure-function relationships of PSMαs and their short truncations. Our results illuminate the formidable structural plasticity of the multifunctional PSM family.

## Results

### The biofilm-associated PSMα1 and PSMα4 form cross-β fibrils.
We found that in contrast to PSMα3, which forms cross-α fibrils[10], the homologous and biofilm-associated[14] PSMα1 and PSMα4 form prototypical amyloid fibrils (Fig. 1a, b and Supplementary Fig. 1) which exhibit the cross-β signature in X-ray diffraction patterns (Fig. 1c, d), and bind the amyloid-indicator dye thioflavin T[22] (Supplementary Fig. 2). It was indeed previously shown that, in solution, PSMα1 and PSMα4 transition from α-helices to parallel β-sheets within several days, whereas PSMα3 remains helical[22]. To obtain atomic-level insights into the architecture of PSMα1 and PSMα4 fibrils, we computationally identified their amyloidogenic spine segments (Supplementary Fig. 1), and determined their high-resolution structures by means

of X-ray microcrystallography (Table 1). Segments PSMα1-IIKVIK and PSMα4-IIKIIK, both of which are conserved within naturally occurring PSM truncations[18–20], adopt the canonical cross-β fibril architecture, wherein pairs of β-sheets tightly mate through a dry interface, forming a steric zipper (Fig. 2, Supplementary Fig. 3a). Both structures belong to the class 1 of steric zippers, indicating that parallel β-sheets mate face-to-face in the fibrils formed by the PSMα1-IIKVIK and PSMα4-IIKIIK segments[23]. X-ray diffraction from these fibrils accordingly exhibit the cross-β pattern (Fig. 2c). Proline substitutions in these spine regions abolished fibril formation of the full-length PSMα1 and PSMα4 (Supplementary Fig. 2). Structural indicators of fibril stability of the IIKVIK and IIKIIK segments, i.e., buried surface area and shape complementarity between sheets, calculated from the crystal structures, resemble those of eukaryotic steric-zipper structures (Supplementary Table 1). The similarity between PSMα1 and PSMα4 segments and human pathological amyloids demonstrate that cross-β amyloids are a structural trait shared across species, from bacteria to human.

### PSMα3-LFKFFK displays antibacterial activity.
PSMα3 is toxic to human cells but certain single-point mutations are known to confer antibacterial activity[24,25]. In addition, natural truncations of PSMs, which occur via proteolysis, could lead to gains in antibacterial activities[17–21]. We found that the LFKFFK segment from PSMα3, identified as fibril-forming by computational methods and which indeed forms fibrils (Fig. 3b), shows dose-dependent antibacterial activity against Gram-positive *Micrococcus luteus* and *Staphylococcus hominis*, but is nontoxic to *S. aureus*, the secreting bacterium (Fig. 3a, Supplementary Fig. 4). The observation that the steric-zipper-forming segments PSMα1-IIKVIK and PSMα4-IIKVIK do not elicit antibacterial activity (Fig. 3a, Supplementary Fig. 4) indicates that fibrillation is, in itself, insufficient to confer toxicity. Likewise, toxicity of PSM segments cannot be ascribed to charges of their amino acids, since

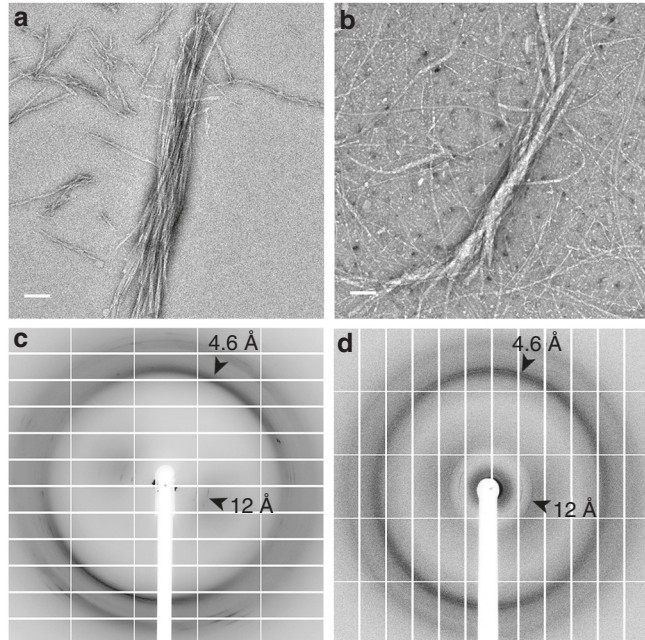

**Fig. 1** PSMα1 and PSMα4 form cross-β amyloid fibrils. Electron micrographs of PSMα1 (**a**) and PSMα4 (**b**) show elongated fibrils; scale bars represent 100 nm. X-ray fibril diffraction of PSMα1 (**c**) and PSMα4 (**d**) shows major diffraction orthogonal arches at 4.6 Å and 12 Å spacings, indicative of the cross-β signature[11]

**Table 1 Data collection and refinement statistics (molecular replacement)**

| | IIKVIK (PSMα1) | IIKIIK (PSMα4) | LFKFFK polymorph I (PSMα3) | LFKFFK polymorph II (PSMα3) |
|---|---|---|---|---|
| PDB accession code | 6FG4 | 6FGR | 6FHC | 6FHD |
| Beamline | ESRF ID23-2 | ESRF ID23-2 | ESRF ID23-2 | EMBL P14 PETRA III |
| Date | October 8th, 2014 | October 8th, 2014 | July 24th, 2014 | May 2nd, 2016 |
| **Data collection** | | | | |
| Space group | C 1 2 1 | P 1 | P 6 5 | C 1 2 1 |
| Cell dimensions | | | | |
| $a, b, c$ (Å) | 45.27 4.80 22.90 | 4.83 22.38 23.06 | 35.79 35.79 9.63 | 41.03 11.73 24.61 |
| $\alpha, \beta, \gamma$ (°) | 90.00 107.65 90.00 | 107.00 90.01 96.20 | 90.00 90.00 120.00 | 90.00 121.89 90.00 |
| Wavelength (Å) | 0.8729 | 0.8729 | 0.8729 | 0.9763 |
| Resolution (Å) | 21.8-1.1 (1.13-1.10) | 22.1-1.5 (1.56-1.50) | 100-1.5 (1.55-1.5) | 20.9-1.8 (1.96-1.85) |
| [a] R-factor observed (%) | 13.3 (75.2) | 20.6 (62.2) | 8.2 (67) | 28.4 (54.7) |
| [b]$R_{meas}$ (%) | 14.0 (82.4) | 21.7 (69.4) | 13.1 (83.1) | 29.2 (56.3) |
| $I$ / sigma | 10.1 (2.1) | 7.2 (2.4) | 20.6 (2.3) | 9.1 (5.6) |
| Total reflections | 23,104 (876) | 13,816 (770) | 16,288 | 16,102 (2590) |
| Unique reflections | 2070 (136) | 1397 (158) | 1179 (118) | 913 (148) |
| Completeness (%) | 93.9 (80.0) | 95.0 (89.3) | 97.4 (98.3) | 96.3 (91.9) |
| Redundancy | 11.2 (6.4) | 9.9 (4.9) | 13.8 (9.6) | 17.6 (17.5) |
| [c] $CC_{1/2}$ (%) | 99.8 (96.7) | 99.4 (74.4) | 96.5 (97.6) | 99.3 (94.6) |
| **Refinement** | | | | |
| Resolution (Å) | 18.4-1.1 (1.23-1.10) | 18.2-1.5 (1.54-1.50) | 31.0-1.5 (1.55-1.51) | 19.3-1.8 (1.90-1.85) |
| Completeness (%) | 94.0 (83.5) | 95.1 (93.9) | 97.8 (98.9) | 97.3 (98.8) |
| [d] No. reflections | 1863 | 1257 | 1057 | 821 |
| [e] $R_{work}$ (%) | 15.9 (21.2) | 18.3 (26.4) | 11.8 (27.7) | 17.1 (29.7) |
| $R_{free}$ (%) | 19.4 (26.4) | 22.1 (26.1) | 16.2 (21.2) | 18.7 (43.6) |
| No. atoms | 77 | 127 | 86 | 136 |
| Protein | 65 | 102 | 60 | 120 |
| Ligand/ion | 10 | 22 | 21 | 11 |
| Water | 2 | 3 | 5 | 5 |
| $B$-factors | | | | |
| Protein | 8.8 | 7.7 (Chain A) 8.2 (Chain B) | 10.2 | 9.6 (Chain A) 11.3 (Chain B) |
| Ligand/ion | 14.5 ($SO_4$) | 37.3 ($SO_4$) 6.7 (EDO) | 25.1 ($CO_3$) 18.3 (SCN) 23.7 (PG4) 46.5 (Cl) | 29.1 ($SO_4$) 25.9 (Na) |
| Water | 33.6 | 26.9 | 49.9 | 23.5 |
| R.m.s. deviations | | | | |
| Bond lengths (Å) | 0.010 | 0.010 | 0.016 | 0.017 |
| Bond angles (°) | 1.973 | 1.903 | 1.995 | 1.893 |
| Clash score [75] | 0.00 | 0.00 | 6.1 | 0 |
| Molprobity score [75] | 1.33 | 0.5 | 1.33 | 1.20 |
| Molprobity percentile [75] | 85th percentile | 100th percentile | 94th percentile | 99th percentile |
| Number of xtals used for scaling | Four spots from one crystal were used. | Four spots from one crystal were used. | One crystal | Three spots from one crystal were used. |

Values in parentheses are for highest-resolution shell
[a]Rfactor square
[b]R-meas is a redundancy-independent R-factor defined in[76]
[c]$CC_{1/2}$ is percentage of correlation between intensities from random half-datasets[77]
[d]Number of reflections corresponds to the working set
[e]Rwork corresponds to working set

all three segments feature two basic and four hydrophobic side chains. Thus, the antibacterial activity of the PSMα3-LFKFFK segment could stem from the specific fibril architecture it adopts.

**LFKFFK forms reversible fibrils and atypical structures.** We obtained two atomic resolution structures of LFKFFK; both revealed a departure from cross-β fibrils and atypical amyloid architectures (Fig. 4). One polymorph was fundamentally different from steric-zippers, displaying no dry interface between pairs of β-sheets. Instead, hexamers of β-sheets formed cylindrical channels running along the fibril-like structure, effectively yielding nanotubes (Fig. 4a). The second polymorph was composed of out-of-register β-sheets (Fig. 4b), meaning that unlike in

canonical cross-β fibrils, β-strands are not perpendicular to the fibril axis[26,27]. Such extreme polymorphism is exceptional within the hundreds of structures of amyloid-like spine segments solved to date[28]. Quantitative measures of amyloid stability based on the crystal structures[28] (Supplementary Table 1) suggest that both polymorphs, and especially the hexameric configuration, form less stable fibrils compared to canonical steric zippers. The hexameric configuration lacks the tight interdigitation of β-sheets characteristic of canonical cross-β fibrils, such as PSMα4-IIKVIK and eukaryotic steric-zipper amyloid segments, resulting in a smaller solvent-accessible surface area buried and lower shape complementarity at the interface between the β-sheets (Supplementary Table 1). The second polymorph, featuring out-of-register β-sheets, shows intermediate quantitative measures of

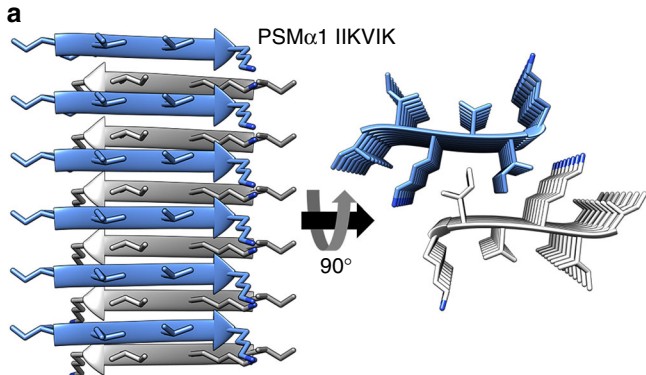

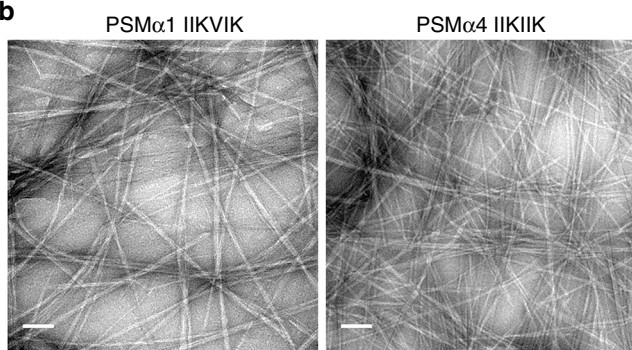

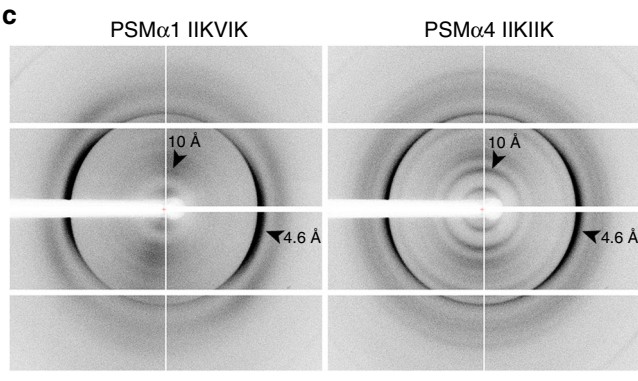

**Fig. 2** PSMα1 and PSMα4 spine segments form steric zipper fibrils.
**a** Crystal structure of the PSMα1 spine segment IIKVIK determined at 1.1 Å resolution, showing the canonical steric-zipper architecture of tightly mated β-sheets composed of parallel β-strands. In the left panel, the view is perpendicular to the fibril axis and the β-strands run horizontally. In the right panel, the view is down the fibril axis. The segments are shown in ribbon representation, with the side chains shown as sticks. The carbons within each β-sheet are colored either gray or light blue, and nitrogen atoms in side chains are colored blue. Here, six layers of β-strands are presented. Theoretically, fibrils can contain thousands of layers. A similar structure of the PSMα4-IIKIIK and its crystal packing are shown in Supplementary Fig. 3. **b** Electron micrographs visualizing PSMα1-IIKVIK and PSMα4-IIKIIK fibrils; scale bars represent 100 nm. **c** X-ray fibril diffraction of PSMα1-IIKVIK and PSMα4-IIKIIK shows major diffraction orthogonal arches at 4.6 Å and 10 Å spacings, indicative of the cross-β signature[11]

amyloid stability, consistent with the previous suggestion that this mating configuration could be associated to the transient nature of amyloid oligomers[26,29]. The contrasting levels of stability of PSMα3-LFKFFK and PSMα4-IIKIIK fibrils, inferred from structural analyses, were verified experimentally. Specifically, PSMα3-LFKFFK fibrils dissolve upon heating to 50 °C, and reform upon cooling, whereas PSMα4-IIKIIK fibrils remain stable (Fig. 5). This

behavior is reminiscent of the reversible fibril formation displayed by low-complexity protein segments associated with membrane-less assemblies, which form fibrils with kinked β-sheets[30,31], and by the TAR DNA-binding protein 43 (TDP-43) that can aggregate both in reversible stress granules and in irreversible pathogenic amyloid[32]. Similar to these human functional amyloids, the labile fibril formation by PSMα3-LFKFFK could underlie a functional role.

**LFKFFK and KLFKFFK share fibril properties and function**. The antibacterial activity of LFKFFK is preserved in the one-residue longer PSMα3 segment KLFKFFK (Fig. 3a, Supplementary Fig. 4). LFKFFK and KLFKFFK both form polymorphic fibrous structures (Fig. 3b), and the secondary structure of their fibrils analyzed using attenuated total-internal reflection Fourier transform infrared (ATR-FTIR) spectroscopy showed similar spectra with the presences of β-rich species (Supplementary Fig. 5). Specifically, the steric-zipper segment PSMα1-IIKVIK showed a peak at 1621 cm$^{-1}$ corresponding to rigid cross-β amyloid fibrils[33–35], in accordance with the crystal structure (Fig. 2). Contrastingly, PSMα3-LFKFFK displayed two main peaks at 1622 cm$^{-1}$ and 1633 cm$^{-1}$, and PSMα3-KLFKFFK displayed a peak at 1633 cm$^{-1}$. The latter is typical of bent β-sheets in protein structures[33–35] and was associated to disorder within amyloid fibrils[33–35], in accordance with the atypical and polymorphic β-rich crystal structures of LFKFFK (Fig. 4). We propose that the antibacterial activities of LFKFFK and KLFKFFK fibrils are encoded in their unique structural properties, including their disordered and polymorphic nature.

## Discussion

Work on human disease-associated amyloids has shown that the vast majority of amyloids are β-rich and polymorphic by nature[28], which was suggested to encode different level of neurotoxicity and prion strains[36]. We hypothesize that functional amyloids are even richer in polymorphisms, as they encode highly diverse functions in a reduced number of related sequences. In the present study, we found considerable structural diversity among close homologs (Supplementary Fig. 6). While PSMα3 forms cross-α fibrils and enhances toxicity against human cells[10], PSMα1 and PSMα4 form cross-β fibrils (Fig. 1) which likely play a role in stabilizing the biofilm matrix[14]. Biofilms formed by pathogenic *S. aureus* strains displaying robust amyloid fibril formation indeed contained a high level of PSMα1 and PSMα4 fibrils[14,37]. The role of amyloid in biofilm development is evidenced in other β-rich microbial amyloids, e.g., curli CsgA from *Escherichia coli*[7] and FabC in *pseudomonas*[38]. The highly stable steric zipper structures (Fig. 2, Supplementary Fig. 3), forming the spines of the cross-β fibrils, putatively serve as the building block cementing the biofilm and creating the rigidity that can explain the resistance of amyloid-containing biofilms[8,14].

A noticeable difference between the steric-zipper structures of the PSMα1 and PSMα4 segments, and the two polymorphs of the LFKFFK segment, is the orientation of the β-strands, namely parallel vs. antiparallel, respectively (Figs. 2 and 4). Fibrils with β-sheets arranged in an antiparallel orientation are generally more toxic than those with a parallel arrangement[39–42]. Moreover, the cylindrical architecture of LFKFFK polymorph I (Fig. 4a) is reminiscent of that displayed by a polymorph of amyloid-β fibrils[43], and a HET-s prion fragment[44]. Similarly, stable toxic oligomeric species of α-synuclein[45] and a segment of the human αB-crystalline, named cylindrin[29] display antiparallel β-strand orientation and cylindrical architectures. The second polymorph of LFKFFK is composed of out-of-register β-sheets (Fig. 4b), a morphology that was also suggested to serve as a pathway to toxic

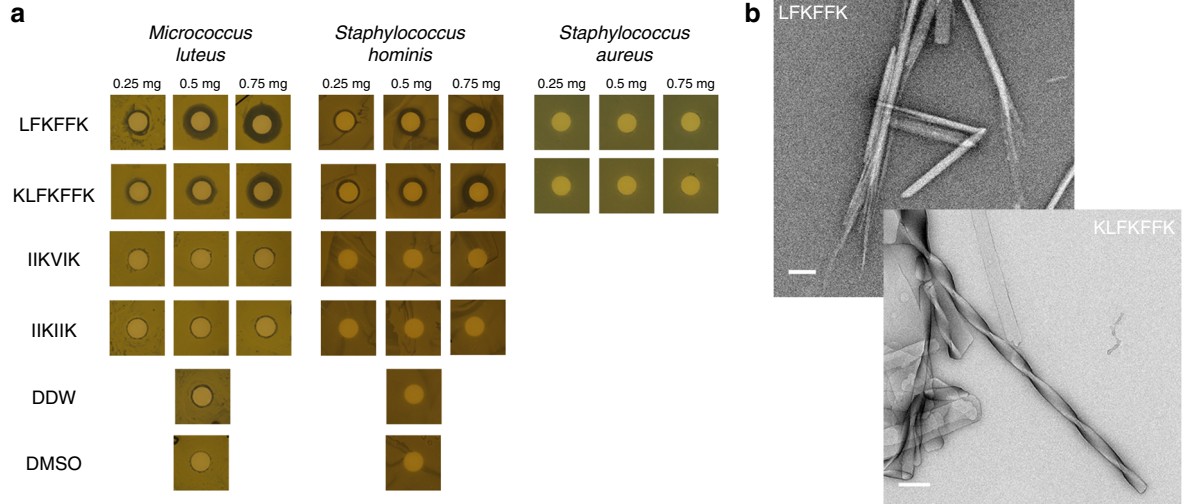

**Fig. 3** Antibacterial activity of the fibril-forming LFKFFK from *S. aureus* PSMα3. Disc diffusion assay testing antibacterial activity against different bacteria. In this assay, the antibacterial agent diffuses into the agar and inhibits germination and growth of the test microorganism. **a** LFKFFK and KLFKFFK segments from PSMα3, but not the steric-zipper forming segments PSMα1-IIKVIK and PSMα4-IIKIIK, showed dose-dependent antibacterial activity against *M. luteus* and *S. hominis*. LFKFFK and KLFKFFK were not toxic to *S. aureus*, the bacterium which secretes PSMα3. Discs soaked with only DDW or Dimethyl-sulfoxide (DMSO) served as controls. The results of the disc diffusion assay coincide with the antibacterial activity seen in solution (Supplementary Fig. 4). **b** Electron micrographs visualizing fibrils and nano-crystals formed by LFKFFK; scale bar represents 400 nm, and straight and twisted crystalline fibrils of KLFKFFK; scale bar represents 200 nm

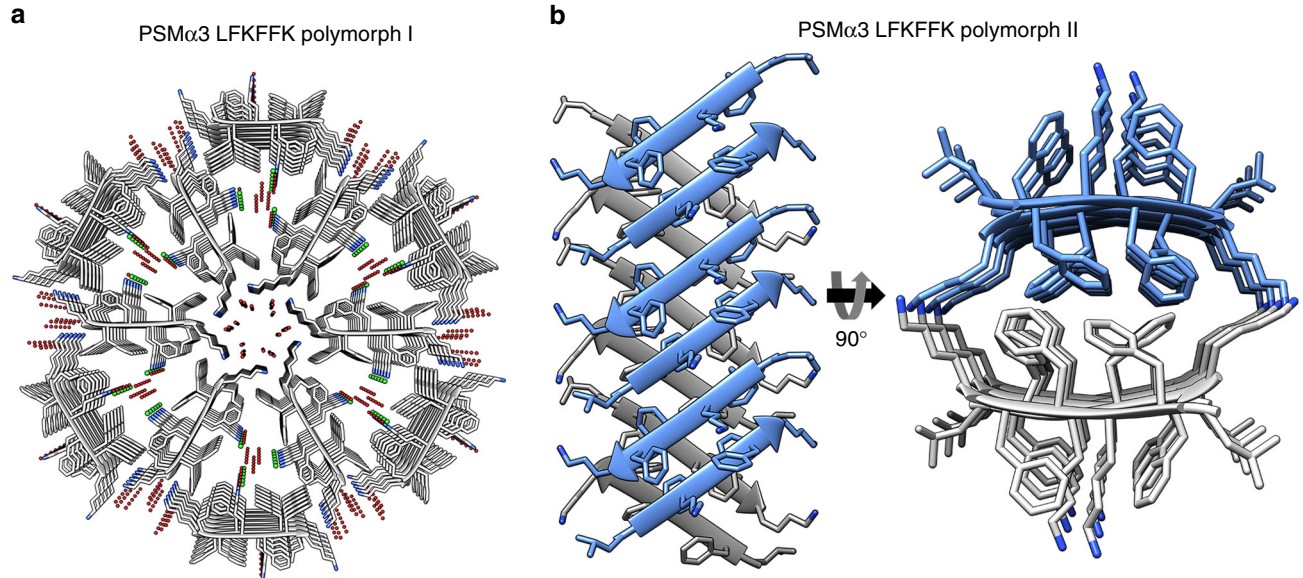

**Fig. 4** Structural polymorphism of the LFKFFK segment from *S. aureus* PSMα3. **a** Crystal structure of polymorph I of the PSMα3 spine segment LFKFFK determined at 1.5 Å resolution. The structure reveals a unique amyloid-like hexameric architecture, which forms elongated cylindrical cavities along the fibril axis. The view is down the fibril axis. The segments are shown in ribbon representation, with side chains shown as sticks with gray carbons and blue nitrogen atoms. Water molecules (oxygen in red) and chloride ions (green) that counteract the charge of the lysine side chains, are shown as small spheres. **b** Crystal structure of LFKFFK polymorph II determined at 1.85 Å resolution, revealed a rare amyloid-like architecture of out-of-register β-sheets, in which each β-strand is at an angle of ~50° from the fibril axis, instead of the close to 90° angle found for in-register sheets. In both polymorphs, the β-sheets are composed of anti-parallel strands. In the left panel, the view is perpendicular to the fibril axis, and in the right panel, the view is down the fibril axis. The segments are shown in ribbon representation, with side chain shown as sticks. The carbons within each β-sheet are colored either gray or light blue, and nitrogen atoms in side chains are colored blue

amyloid aggregates[26], and the interface between the sheets is reminiscent of the KLVFFA segment of amyloid-β[42]. Thus, the two different polymorphs of the antibacterial peptide LFKFFK display features which, in human disease-associated amyloids, correlate with toxicity. Classical steric-zippers and cross-β mature fibrils of amyloids are considered to lack the neurotoxicity that

has been attributed to smaller, transient, oligomers[46]. This points to transient, less stable and often reversible species with self-assembly properties as the toxic entity in pathological amyloids. We correspondingly suggest that atypical fibrils of LFKFFK encode toxicity to bacteria, while the reversible fibril formation provides means to regulate activity.

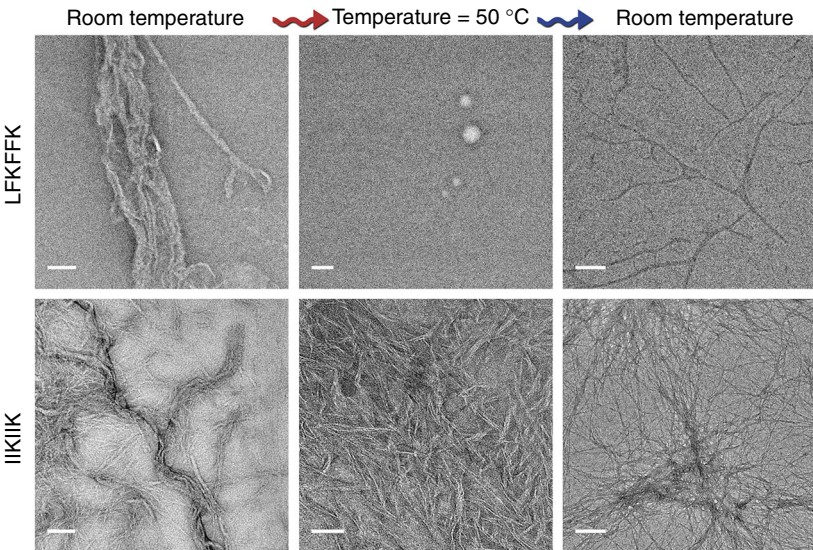

**Fig. 5** LFKFFK forms thermo-reversible fibrils compared to the stable IIKIIK fibrils. LFKFFK formed fibrils at room temperature, which dissolved upon heating to 50 °C and reformed after cooling, albeit with a different morphology; scale bars represent 500 nm for the sample at room temperature and 300 nm for the samples taken after heating to 50 °C and after re-cooling. In contrast, IIKIIK fibrils were thermo-stable; scale bars represent 100 nm for the sample at room temperature, 50 nm for the sample taken after heating to 50 °C, and 300 nm for the sample taken after re-cooling. Both peptides were incubated for 3 days at room temperature before fixation and heating-cooling cycles. The same sample was assayed following temperature change. Thermostability of the fibrils corresponded to the crystal structures; IIKIIK showed the highly stable steric-zipper architecture, while LFKFFK formed atypical amyloid-like structures

The PSMα3 sequence offers a fascinating peek at the complexity of the amyloid fold, populating β-rich amyloid states in the truncated antibacterial LFKFFK form, but forming cross-α fibril in the context of cytotoxic full-length PSMα3[10]. By not representing a spine that recapitulates the properties of the full-length PSMα3, the LFKFFK segment exemplifies how an array of activities can be derived from a single polypeptide, through generation of shorter derivatives with different structural properties. This proposal is in agreement with previous observations that PSMs undergo truncations in vivo via proteolysis, yielding derivatives endowed with new functions such as antibacterial activities[17–21]. The biological relevance of the LFKFFK segment is supported by the observation that LFKFFK is nontoxic to the PSMα3-secreting *S. aureus*, but toxic to the closely related *S. hominis* and to *M. luteus* (Fig. 3). This suggests a mechanism by which *S. aureus* is able to adapt to the toxic fold of LFKFFK fibrils.

A link between amyloids and antibacterial activity has been previously proposed, following the observations that several human host defense peptides form amyloid-like fibrils[47,48], and that several human pathological amyloids display antimicrobial action[49–52]. LFKFFK presents an additional example. Moreover, LFKFFK associates antibacterial activity with atypical and reversible fibril architectures, which differ from those yielding the highly stable fibrils playing a structural role in biofilms. Remarkably, the shortest recognized peptide that can facilitate antimicrobial activity by molecular self-assembly is diphenylalanine (FF)[53], which was first identified as the core recognition module of amyloid-β involved in Alzheimer's disease[54]. FF forms stiff nanotubes[54,55], and its crystal structure revealed channels formed via hexagonal symmetry forming elongated tubes[56,57]. The FF motif is present in the LFKFFK segment, and the hexameric structure of LFKFFK resembles the FF nanotubes. This farther support the role of self-assembly in antibacterial activity and specifically the hexamric arrangement that forms channels along the fibril axis.

Together we found that PSMαs demonstrate a vast structural diversity of amyloid-like structures, including cross-α, cross-β,

out-of-register β-sheets, and hexameric configurations (Supplementary Fig. 6). By simultaneously secreting PSMαs located at the same operon, and by truncations of these peptides, the *S. aureus* bacterium generates various virulent activities encoded by diverse amyloid morphologies. PSMs have critical and diverse roles during infection and represent a promising target for anti-staphylococcal therapy[15]. The atomic structures of the PSMα1 and PSMα4 derivatives offer templates for the design of anti-biofilm compounds. In addition, the antibacterial activity of LFKFFK and KLFKFFK may be advantageous in the development of antimicrobial peptides based on amyloid segments.

## Methods

**Peptides and reagents**. PSM native peptides and shorter derivatives (custom synthesis) at >98% purity (see Table 2) were purchased from GL Biochem (Shanghai) Ltd., as well as from GenScript. The short (6–7 residues) PSMα segments were synthesized with capped termini (acetylated in the N-terminus and amidated in the C-terminus) or with unmodified termini for crystallography. PSMα full-length peptides were synthesized with unmodified termini. Hexafluoroisopropanol (HFIP) and Thioflavin T (ThT) were purchased from Sigma-Aldrich. Ultra-pure water was purchased from Biological Industries. Dimethyl-sulfoxide (DMSO) was purchased from Merck.

PSMαs wild type (WT) and mutant peptide sequences (Uniprot accession codes are in parentheses); segments of short derivatives used here are marked in bold; introduced proline substitutions are underlined.

**Peptides pre-treatment**. Lyophilized synthetic PSMαs were dissolved in HFIP to a concentration of 0.5 mg ml$^{-1}$ followed by a 10 min sonication in a bath-sonicator at room temperature. The organic solvent was evaporated using a mini rotational vacuum concentrator (Christ, Germany) at 1000 rpm for 2 h at room temperature. Treated peptides were aliquoted and stored at −20 °C prior to use.

**Computational prediction of amyloid spine segments**. Amyloidogenic propensities of segments from PSMαs were predicted using combined information from several computational methods, including ZipperDB[58], Tango[59,60], Waltz[61], and Zyggregator[62].

**Fibril X-ray diffraction**. Pre-treated PSMα1 and PSMα4 peptide aliquots were re-dissolved in DMSO to 20 mM and immediately diluted 2-fold in ultra-pure water. Short spine segments with capped termini (IIKVIK from PSMα1, IIKIIK from PSMα4) were re-dissolved to 20 mg ml$^{-1}$ in ultra-pure water. To prepare the sample, 2 μl droplets of the peptide solution were placed between two sealed-end

**Table 2 PSMαs wild type (WT) and mutant peptide sequences (Uniprot accession codes are in parenthesis)**

| Peptide name | Sequence |
|---|---|
| PSM α1 (H9BRQ5) | MGIIAG**IIKVIK**SLIEQFTGK |
| PSM α1 mutant I7P/K9P | MGIIAGPIPVIKSLIEQFTGK |
| PSM α3 (H9BRQ7) | MEFVAK**LFKFFK**DLLGKFLGNN |
| PSM α4 (H9BRQ8) | MAIVGT**IIKIIK**AIIDIFAK |
| PSM α4 mutant I8P/I10P | MAIVGTIPKPIKAIIDIFAK |

Segments of short derivatives used here are marked in bold and introduced proline substitutions are underlined

glass capillaries. PSMα1 and PSMα4 were incubated at 37 °C for 2 days to allow fibril formation. IIKVIK and IIKIIK were incubated at room temperature until the drop dried completely. X-ray diffraction of PSMα1 fibrils was collected at the micro-focus beamline P14 operated by EMBL Hamburg at the PETRA III storage ring (DESY, Hamburg, Germany). X-ray diffraction of PSMα4 fibrils was collected at the micro-focus beamline ID23-2 of the European Synchrotron Radiation Facility (ESRF) in Grenoble, France. X-ray diffraction data from fibrils of IIKVIK and IIKIIK was collected at the micro-focus beamline MASSIF-3 (ID30A-3) of ESRF, Grenoble, France.

**Thioflavin T fluorescence fibrillation kinetics assay.** Thioflavin T (ThT) is a widely used "gold standard" stain for identifying and exploring formation kinetics of amyloid fibrils, both in vivo and in vitro[63]. Fibrillation curves in presence of ThT commonly show a lag time for the nucleation step, followed by rapid aggregation. Pre-treated PSMα1, PSMα1 I7P/K9P, PSMα4, and PSMα4 I8P/I10P peptide aliquots were dissolved in DMSO to 10 mM and immediately diluted to 50 μM in Tris buffer pH 7.5 for PSMα1 and PSMα1 I7P/K9P, or in ultra-pure water for PSMα4 and PSMα4 I8P/I10P, containing filtered ThT diluted from stock made in ultra-pure water. Final concentrations for each reaction were 50 μM peptide (PSMα1, PSMα4, or their mutants) and 200 μM ThT. Blank solutions were also prepared for each reaction, containing everything but the peptides. The reaction mixture was carried out in a black 96-well flat-bottom plate (Greiner bio-one) covered with a thermal seal film (EXCEL scientific) and incubated in a plate reader (CLAR-IOstar BMG LABTECH) at a temperature of 37 °C with 500 rpm shaking for 85 s before each reading cycle, and up to 1000 cycles of 6 min each. Measurements were made in triplicates. Fluorescence was measured by excitation at 438 ± 20 nm and emission at 490 ± 20 nm over a period of about 100 h. All triplicate values were averaged, appropriate blanks were subtracted, and the resulting values were plotted against time. Calculated standard errors of the mean are presented as error bars. The entire experiment was repeated at least three times on different days.

**Transmission electron microscopy.** Fibrillated PSMα1 and PSMα4 samples were collected following the ThT fibrillation kinetics assay (as described above) by combining the contents of 2–3 wells from the plate. Solutions were centrifuged at $21,000 \times g$ and the supernatant was discarded. The pellet containing the fibrils was re-suspended in 20 μl of ultra-pure water.

Fibril formation of the short segments, IIKIIK, IIKVIK, LFKFFK and KLFKFFK, was examined for peptides dissolved directly from the powder form. IIKVIK from PSMα1 and IIKIIK from PSMα4 were re-suspended to 20 mM with 75% DMSO in ultra-pure water and incubated for about 1 week at room temperature (Fig. 2b). LFKFFK and KLFKFFK from PSMα3 were re-suspended in ultra-pure water to 10 mM. LFKFFK was incubated for about a month and KLFKFFK was incubated for 14 days at room temperature (Fig. 3b).

Five-microliter samples were applied directly onto copper TEM grids with support films of Formvar/Carbon (Ted Pella), which were charged by glow-discharge (PELCO easiGlow, Ted Pella) immediately before use. Grids were allowed to adhere for 2 min and negatively stained with 5 μl of 2% uranyl acetate solution. Micrographs were recorded using FEI Tecnai G2 T20 S-Twin transmission electron microscope operated at an accelerating voltage of 200 KeV (located at the Department of Materials Science & Engineering at the Technion, Israel), FEI Tecnai G2 T12 TEM operated at an accelerating voltage of 120 kV (located at the Russell Berrie Electron Microscopy Center of Soft Matter at the Technion, Israel). Images were recorded digitally by a Gatan US 1000 CCD camera using the DigitalMicrograph® software.

**Fibril thermal-stability assay.** For the fibril thermal-stability assay (Fig. 5), peptides were dissolved to 10 mM (LFKFFK in ultra-pure water and IIKIIK in DMSO) and incubated at room temperature for 3 days. Samples were applied directly onto copper TEM grids and stained as described above. The samples were then incubated at 50 °C in a pre-heated thermoblock for 5 min and immediately applied onto TEM grids. The samples were then allowed to cool down at room temperature for 20 min and applied onto TEM grids. Micrographs were recorded as described above.

**Secondary structure analysis using ATR-FTIR spectroscopy.** The short spine segments (IIKVIK from PSMα1, LFKFFK and KLFKFFK from PSMα3) were dissolved to 1 mg ml$^{-1}$ in 5 mM hydrochloric acid (HCl) and sonicated at a bath sonicator for 5 min at room temperature. The solutions were frozen in liquid nitrogen and lyophilized over-night to complete dryness. The procedure was repeated three times to completely remove Trifluoroacetic acid (TFA) residues from peptide synthesis, as TFA has a strong FTIR signal. The dry peptide was then dissolved in D$_2$O to 20 mg ml$^{-1}$ for IIKVIK, and 40 mg ml$^{-1}$ for LFKFFK and KLFKFFK immediately prior to measurements. Five microliters of each peptide solution were spread on the surface of the ATR module (MIRacle Diamond w/ZnSe lens 3-Reflection HATR Plate; Pike Technologies) and let dry under nitrogen gas to purge water vapors. Absorption spectra were recorded on the dry samples using Tensor 27 FTIR spectrometer (Bruker Optics). Measurements were performed in the wavelength range of 1500–1800 cm$^{-1}$ in 2 cm$^{-1}$ steps and averaged over 100 scans. Background (ATR module) and blank (D$_2$O) were measured and subtracted from the final spectra. The amide I' region of the spectra (1600–1700 cm$^{-1}$) is presented (Supplementary Fig. 5).

**Crystallizing conditions.** Peptides synthesized with free (unmodified) termini were used for crystallization experiments. IIKVIK of PSMα1 and IIKIIK of PSMα4 were dissolved in 0.5% acetonitrile in ultra-pure water at 10 mM, and LFKFFK of PSMα3 was dissolved in ultra-pure water at 10 mM, followed by a 10 min sonication in a bath-sonicator at room temperature and centrifugation at $17,000 \times g$ at 4 °C prior to crystallization. Peptide solution drops at 10 nl volume were dispensed by the Mosquito automated liquid dispensing robot (TTP Labtech, UK), to crystallization screening plates. All crystals were grown at 20 °C via hanging-drop vapor diffusion. The drop was a mixture of 10 mM peptide and reservoir solution as follows: IIKVIK: 0.5 M Lithium sulfate; 15% polyethylene glycol 8000. IIKIIK: 0.2 M Ammonium sulfate; 20% polyethylene glycol 3350. LFKFFK polymorph I: 0.2 M Potassium thiocyanate; 20% polyethylene glycol 3350. LFKFFK polymorph II: 0.1 M Sodium acetate pH 5.1; 45% polyethylene glycol 400; 0.09 M Lithium sulfate. Micro-crystals grew after a few days and were mounted on glass needles glued to brass pins[64]. Crystals were kept at room temperature prior to data collection.

**Structure determination and refinement.** X-ray diffraction data was collected at 100 K. The X-ray diffraction data for IIKVIK, IIKIIK, and LFKFFK polymorph I was collected at the micro-focus beamline ID23-EH2 of the European Synchrotron Radiation Facility (ESRF) in Grenoble, France; wavelength of data collection was 0.8729 Å. The X-ray diffraction data for LFKFFK polymorph II was collected at the micro-focus beamline P14 operated by EMBL at the PETRAIII storage ring (DESY, Hamburg, Germany); wavelength of data collection was 0.9763 Å. Data indexation, integration and scaling were performed using XDS/XSCALE[65] or with DENZO/SCALEPACK[66]. Molecular replacement solutions for all segments were obtained using the program Phaser[67] within the CCP4 suite[67,68]. The search models consisted of available structures of geometrically idealized β-strands. Crystallographic refinements were performed with the program Refmac5[69]. Model building was performed with Coot[70] and illustrated with Chimera[71]. There were no residues that fell in the disallowed region of the Ramachandran plot. Crystallographic statistics are listed in Table 1.

**Calculations of structural properties.** The Lawrence and Colman's shape complementarity index[72] was used to calculate the shape complementarity between pairs of sheets forming the dry interface. Area buried was calculated using AREAIMOL[73,74] with a probe radius of 1.4 Å. Calculations were performed using the CCP4 package[68]. The summation of the differences between the accessible surface areas of one molecule alone and in contact with the other strands on the same sheet or opposite sheets, as indicated in Supplementary Table 1, constitutes the reported area buried.

**Bacterial strains and culture media.** *Micrococcus luteus* (an environmental isolate) was a kind gift from Prof. Charles Greenblatt from the Hebrew university in Jerusalem, Israel. An inoculum was grown in Luria-Bertani (LB) medium at 30 °C with 220 rpm shaking overnight. *Staphylococcus hominis* (ATTC 27844) and *Staphylococcus aureus* (ATCC 25923) were purchased from ATCC, USA. An inoculum was grown in brain-heart infusion (BHI) media at 37 °C with 220 rpm shaking overnight.

**Disc diffusion assay.** Bacterial cultures were grown overnight as described above and diluted 1000-fold to a fresh media until growth reached to optical density of ~0.4 measured at 600 nm (OD$_{600}$). The refreshed cultures were plated on LB-agar or BHI-agar according to the bacterial strain. The antibacterial activity of PSMα segments was examined: Lyophilized peptides were dissolved in ultra-pure water (LFKFFK and KLFKFFK) or DMSO (IIKVIK and IIKIIK) to a final concentration of 50 mg ml$^{-1}$. The concentrated peptide solutions were loaded on blank antimicrobial susceptibility discs (Oxoid, UK). Discs loaded with ultra-pure water or DMSO were used as controls. The discs were gently placed on bacteria plated agar and incubated over-night at the appropriate temperatures.

**Antibacterial activity in solution**. Bacterial cultures were grown overnight as described above and diluted 1000-fold to a fresh media until growth reached to $OD_{600}$ of ~0.4. Peptides were dissolved to a concentration of 10 mM (LFKFFK and KLFKFFK in ultra-pure water, IIKVIK and IIKIIK in DMSO) and diluted to 1 mM in growth media (LB for *M. luteus* and BHI for *S. hominis*). Two-fold serial dilutions of the tested peptides in LB or BHI media ranging from 250 μM to 1 μM were performed in a sterile 96-well plate. Final DMSO concentration was fixed to 2.5% in all samples. Wells containing everything but the peptide served as controls. Bacterial growth was determined by measuring the $OD_{595}$ during a 24 h incubation at 30°/37 °C according to the bacterial strain with 250 rpm shaking in a plate reader (FLUOstar omega or CLARIOstar, BMG LABTECH). The experiment was performed in triplicates. All triplicate values were averaged, appropriate blanks were subtracted, and the resulting values were plotted against peptide concentration. Calculated standard errors of the mean are presented as error bars. The entire experiment was repeated at least three times on different days.

## Data availability

The data that support the findings of this study are available on request from the corresponding author. Coordinates and structure factors for the X-ray crystal structures have been deposited in the protein data bank (PDB) with accession codes 6FG4 (IIKVIK), 6FGR (IIKIIK), 6FHC (LFKFFK polymorph I) and 6FHD (LFKFFK polymorph II).

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

## Acknowledgements

We are grateful to M. Sawaya, N. Ben-Tal, M. Chapman, N. Jain, M. Evans, A. Syed, Z. Hayouka, H. Bochnik-Tamir, O. Tabachnikov, T. Gera, E. Tamar, I. Yelin, O. Snitser, and R. Edrei for help with experiments and fruitful discussions. We acknowledge Y. Pazy-Benhar and D. Hiya at the Technion Center for Structural Biology (TCSB), and the Electron Microscopy Center located at the Technion's Department of Materials Science & Engineering, and the Russell Berrie Electron Microscopy Center of Soft Matter. This research was supported by the I-CORE Program of the Planning and Budgeting Committee and The Israel Science Foundation, Center of Excellence in Integrated Structural Cell Biology (grant no. 1775/12), DFG: Deutsch-IsraelischeProjektkooperation (DIP) (Grant No. LA 3655/1-1), Israel Science Foundation (grant no. 560/16), University of Michigan—Israel Collaborative Research Grant, and BioStruct-X, funded by FP7. J.-P. C. acknowledges financial support by CEA, CNRS, Université Grenoble Alpes, and the Agence Nationale de la Recherche (Grant No. ANR-15-CE18-0005-02). The synchrotron MX data collection experiments were performed at beamlines ID23-EH2 and MASSIF-3 (ID30A-3) at the European Synchrotron Radiation Facility (ESRF), Grenoble, France, and at beamline P14, operated by EMBL Hamburg at the PETRA III storage ring (DESY, Hamburg, Germany). We are grateful to the teams at ESRF and EMBL Hamburg.

## Author contributions

N.S. and M.L. conceived the study and designed the experiments. All authors collected X-ray data. A.M. initiated experiments. N.S. performed the experiments. N.S. and M.L. solved the crystal structures with contributions from J-P.C. N.S., J-P.C. and M.L. wrote the paper.

## Additional information

**Competing interests:** The authors declare no competing interests.

