## [Peer Review File · Nature Communications]

Reviewers' comments:

Reviewer #1 (Remarks to the Author):

Salinas Manuscript Review

This work provides high quality structural information corresponding to 6 and 7mer peptides derived from *S. aureus* PSMs. The PSMa1/4 peptides form striking cross- β strand amyloid fibrils with a steric zipper-like structure. The authors show that truncated PSMa3 peptide can adopt structurally reversible polymers of at least 2 forms. The peptides derived from PSMa3 are bacteriostatic against certain Gram-positive bugs. There are a few points the authors should address before the manuscript is recommended for publication.

- Other than Fig. 1 & Supp Fig. 1, all other data shown uses the truncated peptides of the PSMa1/3/4 proteins. The authors should carefully parse their claims and not assign properties of the truncated peptides to the full-length proteins, unless experimentally demonstrated. The focus of the data is with the 6/7mer peptides, not with full length PSMs.
- The authors provide little evidence to support the notion that these new amyloid morphologies confer specific levels of structural integrity. They rely on other literature to support that claim.

Minor issues:

- The manuscript should include line numbers for easier editing.
- At the end of the first paragraph in the Results section on pg 5 the authors refer to Supplementary Table 2 which compares the structure of truncated PSMa1/3/4 peptide amyloids with NNQQNY peptide amyloids. The table identifies NNQQNY peptide as the yeast prion amyloidogenic sequence. Therefore, the claim of the trait being shared "from bacteria to human" seems unsupported.
- In Supplemental Figure 2, when does the PSMa4 plateau? At 100 hrs the ThT fluorescence continues to rise. Do the authors care to comment on this relatively slow polymerization kinetics?
- Supp Fig. 5 is a beautiful electron micrograph. However, I am not sure how it fits into the narrative of the manuscript. Can the authors elaborate on what the twisted crystalline morphology might mean? Specifically, it would be interesting for the authors to expand on their thoughts started in the last sentence on Page 5, where the authors point to Supp Fig. 5 to support the claim that "antibiotic activity is preserved"

Reviewer #2 (Remarks to the Author):

The Manuscript by Salinas et al. describes the crystal structures arising from fragments homologous to regions of the *Staph. Aureus* proteins "PSMalpha" peptides. The authors have previously shown that PSMa-3 forms cross-alpha arrangements, whilst here they show that PSMa-1 and 4 form cross-beta structures. Sections from these peptides were then crystallised and their structures solved revealing a series of interesting structures. The structures of IIKVIK and IIKIHK from PSMa1 and 4 are cross-beta spines and neither of these peptides cause toxicity to bacteria. Interestingly, LFKFFK and KLFFFK are toxic. Also, these fibrils are unstable and can be disassembled. The crystal structures show that LFKFFK forms two different interesting polymorphs, one which is a cylinder and the other, of 50° tilted sheets.

It is interesting that full length PSMa forms cross alpha, while the segment forms beta sheet polymorphs. The structures are all very interesting and clear.

I have some comments

- 1). What does the fibre diffraction pattern from LFKFFK look like? Does it appear as "cross-beta" or does it show different reflections arising from these unusual crystal structures.
- 2) the observation that PSMa-3 full length is cross-a, whilst the segment is cross-b begs the important question about how relevant these crystal structures of short peptides are to the full-

length form. I am concerned that the segments will form amyloid fibrils (cross-beta) BUT these do not impact on the structure for the full length peptide. Therefore, how much can we conclude about the function from these studies?

3) Could the differences in toxic effects arise from the stability of the assemblies rather than their final structures. Indeed, would we expect these novel beta structures formed by LFKFFK to give rise to the reversibility, while classical cross-beta zippers would be more stable. Could this explain the differences? More discussion of this would be very important here.

4. In the introduction, the authors refer to the PSMa-3 structure as amyloid. The original paper refers to these filaments as amyloid. However, the structure does not appear to correlate with the accepted nomenclature for amyloid, as described by the Amyloid nomenclature committee.

<https://www.tandfonline.com/doi/full/10.3109/13506129.2014.964858>

The word amyloid is unnecessary here when describing these cross-alpha structures and should be removed to avoid confusion.

Reviewer #3 (Remarks to the Author):

Overview:

The article by Salinas et al. describes the amyloid assemblies formed by segments derived from the phenol-soluble modulin peptide family (PSMs). This work follows another article by the same group of authors on the structure of PSM-alpha3, a cytotoxic peptide. That structure of the 22-residue peptide revealed a tightly packed alpha helix with amyloid-like properties and was published in Science.

Through the use of x-ray micro diffraction, the group now reveals additional structures of potentially functional amyloids formed by PSM segments. The segments in question are overall shorter than that previously described for PSM-alpha3, and one that is a fragment of that previously published structure. This article is of broad interest to the microbial amyloid community, revealing the variety of polymorphs potentially formed by bacterial amyloids.

The article raises technical and scientific questions, detailed below that could be addressed by minor revision.

Questions/comments:

The 'extreme' polymorphism referred to in the title appears only to apply to the segments associated with PSM-alpha3. In fact those of PSM-alpha1/4 are nearly identical in structure - by eye, the C-alpha RMSD appears near zero (not surprising given the high sequence identity between the two segments). This discordance plays two different stories to the reader, one in which two different PSMs look near identical, and the other where a single PSM is a shape-shifter.

How do the fibril diffraction patterns shown in Figure 1 for PSM-alpha1/4 compare with simulated or measured fibril diffraction patterns from the PSM-alpha1/4 spine segments whose structures were determined here?

The selection of the two spine segments from PSM-alpha1/4 seems ambiguous. Why not consider the region immediately following these spines (AIIIDIF) which at least for PSM-alpha4 is largely hydrophobic?

How does the structure of KLFKFFK compare to that of its shorter counterpart? Does the polymorphism seen for the shorter PSM-alpha3 segment depend on its length? If so, the evidence provided may not be representative of what is observed in longer, bioactive segments. The single micrograph of KLFKFFK provides little in the form of structural evidence to disambiguate these questions.

What does the comparison in Supplementary Figure 7 offer? Wouldn't we learn more from a comparison of the cross-alpha and cross-beta structures of PSM-alpha3?

The article touches on the labile nature of some of the segments investigated, but does not demonstrate any of the biochemical features associated with reversible amyloid behavior - phase separation, coalescence, etc. Do segments of PSM-alpha3 represent bonafide reversible amyloid? This area merits further investigation; its present description seems tangential and potentially premature.

Response to Reviewers' comments

We appreciate the constructive comments provided by the Reviewers that greatly contributed to the manuscript and clarified the results. Below, please find a point-by-point response to the Reviewers' comments. Main changes made in the manuscript are highlighted in the revised version.

Reviewer #1:

"This work provides high quality structural information corresponding to 6 and 7mer peptides derived from S. aureus PSMs. The PSM α 1/4 peptides form striking cross- β strand amyloid fibrils with a steric zipper-like structure. The authors show that truncated PSM α 3 peptide can adopt structurally reversible polymers of at least 2 forms. The peptides derived from PSM α 3 are bacteriostatic against certain Gram-positive bugs. There are a few points the authors should address before the manuscript is recommended for publication."

1. *"Other than Fig. 1 & Supp Fig. 1, all other data shown uses the truncated peptides of the PSM α 1/3/4 proteins. The authors should carefully parse their claims and not assign properties of the truncated peptides to the full-length proteins, unless experimentally demonstrated. The focus of the data is with the 6/7mer peptides, not with full length PSMs."*

We thank the reviewer for this suggestion. To avoid sowing confusion, we have clarified all referrals to either full-length or truncations.

2. *"The authors provide little evidence to support the notion that these new amyloid morphologies confer specific levels of structural integrity. They rely on other literature to support that claim."*

We are not sure that we understand correctly the Reviewer's comment, thus we hope that we provided the appropriate answer regarding structural integrity:

The canonical steric zipper structures are considered in the literature to be highly stable, which arises from the interdigitated dual sheets¹. Since steric-zipper fibrils are unusual in that pairs of β -sheets mate more closely than the adjoining surfaces in other protein complexes, quantitative measures of amyloid stability are based on solvent-accessible surface area buried at the interface between the mating sheets, which is typically 150–200 Å² per β -strand¹. Our calculations of the properties of the PSM segment structures show that in the steric-zipper structure of IIKIIK and IIKVIK the per-strand buried surface area is ~260 Å². The LFKFFK hexameric polymorph shows much smaller solvent-accessible

surface area buried per strand, $\sim 130 \text{ \AA}^2$, pointing to lower stability. (We note that in Supplementary Table 2 the values are calculated for both sides of the interface, in order to provide more accurate comparison of the total area buried).

An additional useful measure is the shape complementarity (Sc), indicating on the closeness of fit of two protein surfaces². Sc of zero indicates no complementarity of the two surfaces and approaches 1.0 for atomic surfaces that fit perfectly together. For example, in antigen-antibody surfaces, the Sc is around 0.66. Steric zipper dry interfaces display comparable or greater values of Sc , for example 0.86 for NNQQNY (from yeast prion) or 0.92 for GGVVIA (from Amyloid- β), which are among the highest values measured for amyloid spines³. In our case, IIKVIK and IIKIIK show $Sc=0.89$ (Supplementary Table 2), which is considered very high, even for amyloid spines. The hexameric structure of LFKFFK shows lower shape complementarity ($Sc=0.79$).

Overall, we provide in Supplementary Table 2 common quantitative measures of amyloid stability based on the crystal structures. We added some explanations to the Table following the Reviewer's comments. The conclusions from the calculations are that the steric-zipper forming spines IIKVIK and IIKIIK are highly stable, even compared to spines of human pathological amyloids, while the LFKFFK polymorphs, especially the hexameric form, are less stable as reflected by their lower solvent-accessible surface area buried and shape complementarity at the interfaces. **Predictions based on calculations from crystal structures are supported by the experimental data testing the thermostability of the fibrils, revealing that the IIKIIK fibrils are thermostable, whereas those formed by LFKFFK are not.** A similar approach was recently used by both Eisenberg and co-workers and Liu and co-workers to show that fragments of amyloid involved in RNA granules (fibrillation associated with membraneless assemblies) are less thermostable compared to segments that form canonical steric-zippers^{4,5}. **To summarize, we based our conclusions on structural integrity on quantitative measures of amyloid stability calculated from the crystal structures, as well as experimental measurements of thermostability of the fibrils.**

Minor issues:

3. *"The manuscript should include line numbers for easier editing."*

We added line numbers. Thank you for the advice which made the revisions easier.

4. *"At the end of the first paragraph in the Results section on pg 5 the authors refer to Supplementary Table 2 which compares the structure of truncated PSM α 1/3/4 peptide amyloids with NNQQNY peptide amyloids. The table identifies NNQQNY peptide as the yeast prion amyloidogenic sequence. Therefore, the claim of the trait being shared "from bacteria to human" seems unsupported."*

We now added to Supplementary Table 2 calculations for human spine segments: VQIVYK from the tau protein and KLVFFA from Amyloid- β . While dozens of steric zipper structures were determined from spines of human proteins by Eisenberg and co-workers¹, we chose to use NNQQNY from yeast prion Sup35 as it shows one of the highest values of shape complementarity and surface area buried among steric zipper structures³. Actually, fibrils of the longer segment GNNQQNY were examined experimentally and displayed an astonishing resistance to chaotropic solvents including 5% SDS or 4 M urea⁶.

5. *“In Supplemental Figure 2, when does the PSM α 4 plateau? At 100 hrs the ThT fluorescence continues to rise. Do the authors care to comment on this relatively slow polymerization kinetics?”*

ThT kinetic assays are mostly useful to indicate amyloidogenic properties of nucleation (indicated by the lag time), followed by rapid aggregation and elongation period. Yet, we note that ThT assays are very sensitive to pre-treatment of the protein/peptides (for example with HFIP or TFA) and to the conditions used, such that it is very difficult to draw conclusions from the exact lag time, or from the time it takes to plateau, among different proteins. We repeated the ThT assay many time, with similar results showing nucleation and elongation, yet the exact lag and elongation times varied. For example, in the experiment below, it takes PSM α 4 longer time to nucleate (40hr), but it reaches a plateau after 80hr. We note that in TEM micrographs, we observed fibrils of PSM α 1 and PSM α 4 after 2 days, which is rather fast when compared to many other amyloid peptides. We nevertheless indeed noticed that PSM α 1 and PSM α 4 are slower in inducing ThT fluorescence compared to PSM α 3. This might be a result of lower water solubility of PSM α 1 and PSM α 4 compared to PSM α 3, weaker ThT binding, time it takes for ThT binding to reach steady state, or lower polymerization rate. It is impossible to explicitly isolate the parameters that determine ThT fibrillation kinetics.

6. *“Supp Fig. 5 is a beautiful electron micrograph. However, I am not sure how it fits into the narrative of the manuscript. Can the authors elaborate on what the twisted crystalline morphology might mean? Specifically, it would be interesting for the authors to expand on their thoughts started in the last sentence on Page 5, where the authors point to Supp Fig. 5 to support the claim that “antibiotic activity is preserved”*

We thank the reviewer for this comment as we realized that the referral to LFKFFK needs further clarification and experiments.

In our aim to understand the structural properties that induce the antibacterial activity of LFKFFK, we hypothesized that the polymorphic, atypical and reversible nature of the fibrils are important for antibacterial activity and regulation. This is in contrast to the IIKVIK and IIKIIK segments, which contain positively charged and hydrophobic residues, yet form canonical, stable, cross- β fibrils. The other segment that we found to have antibacterial activity is LFKFFK. We could not determine the crystal structure of this segment, but observed that, similarly to LFKFFK, it forms polymorphic crystalline fibrils with some straight and some twisting morphologies (we now display both LFKFFK and LFKFFK micrographs in Fig. 3). We added an analysis of the secondary structure of the fibrils using attenuated total-internal reflection Fourier transform infrared (ATR-FTIR) spectroscopy (lines 144-156), which showed similar spectra indicative of β -rich species for LFKFFK and LFKFFK (Supplementary Fig. 5). Specifically, the steric-zipper segment PSM α 1-IIKVIK shows a peak at 1621 cm^{-1} corresponding to rigid cross- β amyloid fibrils⁷⁻⁹, in accordance with the crystal structure (Fig. 2). Contrastingly, PSM α 3-LFKFFK shows two main peaks at 1622 cm^{-1} and 1633 cm^{-1} and PSM α 3-LFKFFK shows a peak at 1633 cm^{-1} , with the latter indicating on more disordered fibers with absorbance which is typical of the bent β -sheets in proteins⁷⁻⁹, in accordance with the atypical and polymorphic β -rich crystal structures of LFKFFK. The similarity between LFKFFK and LFKFFK fibrils, and their disordered and polymorphic nature compared to steric-zippers, further propose that the unique structural properties of these self-assembling peptides encode their antibacterial activity.

Reviewer #2

“The Manuscript by Salinas et al. describes the crystal structures arising from fragments homologous to regions of the Staph. Aureus proteins “PSMalpha” peptides. The authors have previously shown that PSMa-3 forms cross-alpha arrangements, whilst here they show that PSMa-1 and 4 form cross-beta structures. Sections from these peptides were then crystallised and their structures solved revealing a series of interesting structures. The structures of IIKVIK and IIKIIK from PSMa1 and 4 are cross-beta spines and neither of these peptides cause toxicity to

bacteria. Interestingly, LFKFFK and KLFFFK are toxic. Also, these fibrils are unstable and can be disassembled. The crystal structures show that LFKFFK forms two different interesting polymorphs, one which is a cylinder and the other, of 50° tilted sheets.

It is interesting that full length PSMa forms cross alpha, while the segment forms beta sheet polymorphs. The structures are all very interesting and clear.

I have some comments"

1. "What does the fibre diffraction pattern from LFKFFK look like? Does it appear as "cross-beta" or does it show different reflections arising from these unusual crystal structures. "

The fibril diffraction pattern of both LFKFFK and KLFKFFA indicates mostly on β -rich structures, showing a primary reflection at 4.7 Å spacing. We measured fibril diffraction from many samples and in some we did observe a cross- β pattern (see figure below). We figured that LFKFFK is highly polymorphic in its fibril architectures and that the stable cross- β is likely to be the endpoint of structural transitions. Cross- β is probably dominant in the extreme conditions needed to prepare fibril diffraction samples (dissolving the peptide in high concentration and drying it completely between two sealed glass capillaries). Actually, working with many amyloidogenic polypeptides, it is common to observe cross- β fibril diffraction as the dominant pattern, while when using methods such as FTIR, multiple structural entities are observed (which are difficult to deduce from X-ray fibril diffraction). We accordingly added an analysis of the secondary structure of the fibrils using ATR-FTIR spectroscopy (lines 144-156), which showed similar spectra for LFKFFK and KLFKFFK with the presences of β -rich species (Supplementary Fig. 5). Specifically, the steric-zipper segment PSM α 1-IIKVIK shows a peak at 1621 cm^{-1} corresponding to rigid cross- β amyloid fibrils⁷⁻⁹, in accordance with the crystal structure (Fig. 2). Contrastingly, PSM α 3-LFKFFK shows two main peaks at 1622 cm^{-1} and 1633 cm^{-1} and PSM α 3-KLFKFFK shows a peak at 1633 cm^{-1} , with the latter indicating on more disordered fibers with absorbance which is typical of the bent β -sheets in proteins⁷⁻⁹, in accordance with the atypical and polymorphic β -rich crystal structures of LFKFFK. The similarity between LFKFFK and KLFKFFK fibrils, and their disordered and polymorphic nature compared to steric-zippers, further propose that the unique structural properties of these self-assembling peptides encode their antibacterial activity.

2. *“the observation that PSM α -3 full length is cross-a, whilst the segment is cross-b begs the important question about how relevant these crystal structures of short peptides are to the full-length form. I am concerned that the segments will form amyloid fibrils (cross-beta) BUT these do not impact on the structure for the full length peptide. Therefore, how much can we conclude about the function from these studies?”*

The reviewer’s question reflects long-lasting debates about the reductionist approach of looking on amyloid spine segments. The information obtained from the structures of the segments is obviously limited. However, especially since the polymorphic and partially disordered nature of the full-length amyloids generally precludes atomic resolution structure determination, the spine structures are our best option to reveal in exquisite detail the atomic factors that account for amyloid structure and stability. For canonical amyloids, biophysical methods such as circular dichroism, ATR-FTIR spectroscopy, and fibril diffraction, showed the common transition into β -rich structures, and the cross- β diffraction pattern. In these cases, the spines that formed steric-zipper structures manifested and exposed the details of the cross- β pattern. Many of these short peptides manifested the properties of the full-length amyloids, namely form elongated fibrils, show enhanced nucleation by seeding, and induce ThT fluoresces showing lag time and elongation, such that we can be more confident about the relevancy of the spines. This is also the case for the cross- β forming PSM α 1 and PSM α 4 which show the compatibility between the cross- β diffraction of the full-length polypeptide and the steric-zipper structures of the spines.

In the case of the PSM α 3 LFKFFK derivative, we propose that the segment does not reflect the properties of the full-length, yet expands the array of activities derived from this polypeptide by generating shorter derivatives with different structural properties. In vivo, PSMs are known to undergo truncation (by proteolysis) in response to various external stimuli, yielding truncated PSMs with new functions such as antibacterial activities¹⁰⁻¹⁴. Our research initiated from the hypothesis that such shorter active derivatives may be prone to form a different type of structure compared to the full-length polypeptide, making the array of structural species much larger than the actual number of PSMs. Although there is very limited information regarding the exact derivatives of PSM α 3, we suggest that LFKFFK, by forming atypical β -rich fibrils, represents such an example, in which a truncated derivative possess a new function, namely antibacterial activity. Moreover, our observation that LFKFFK is active against two Gram positive bacterial strains, but not against *S. aureus* (the PSM secreting bacteria), further supports biological relevance. **We added discussions on the relationship between the segment and the full-length in the revised text (lines 193-198).**

We would like to note that recent unpublished results from our lab reveals that single-point mutants of PSM α 3, some showing antibacterial gain-of-function, display mixed helical and β -rich structural properties (Tayeb-Fligelman and Landau, *in preparation*). These results support the notion that while the full-length PSM α 3 is helical in nature, it embeds the ability to form β -rich structures in its sequence, as manifested by its shorter derivatives and single-point mutants (which abrogate the formation of particular salt bridges required to stabilize the helical conformation). We suggest that this structural diversity is used to encode different functions needed in *S. aureus* in different environments and stress conditions. We figure that much more work will be needed for deciphering the full extent of the structure-function-fibrillation relationship in this system.

3. *“Could the differences in toxic effects arise from the stability of the assemblies rather than their final structures. Indeed, would we expect these novel beta structures formed by LFKFFK to give rise to the reversibility, while classical cross-beta zippers would be more stable. Could this explain the differences? More discussion of this would be very important here.”*

We honestly cannot say in certainty that the antibacterial activity is arising from lack of stability or the final structure. Yet, we believe that one or more atypical LFKFFK conformations is toxic to bacteria while fibril reversibility is related to regulation of activity. Classical cross- β mature fibrils of amyloids are considered to lack the neurotoxicity that has been attributed to smaller, transient, oligomers. It indicates that transient, less stable species with self-assembly properties are the toxic entity in canonical amyloids. Thus, the unstable fibril architectures of LFKFFK could serve as the toxic entity. Recent studies on human functional amyloids showed reversible fibril formation of low-complexity protein segments associated with membraneless assemblies, while the structures of the amyloidogenic segments showed fibrils with kinked β -sheets (thus less stable)^{4,5}. In this human functional amyloid, the labile fibril formation underlies regulation of function. We believe that this is the case for LFKFFK. We added this discussion to the revised manuscript (lines 185-190).

4. *“In the introduction, the authors refer to the PSM α -3 structure as amyloid. The original paper refers to these filaments as amyloid. However, the structure does not appear to correlate with the accepted nomenclature for amyloid, as described by the Amyloid nomenclature committee. <https://www.tandfonline.com/doi/full/10.3109/13506129.2014.964858> The word amyloid is unnecessary here when describing these cross-alpha structures and should be removed to avoid confusion.”*

The definition of amyloids is indeed currently restricted to cross- β polymers, yet nomenclature of amyloids was constantly revised over the last decades and is based on what was discovered over the years. Initially, pathologists classified protein fibrils as amyloid if they are associated with disease and if they bind Congo red, displaying a green birefringence. This was obviously a limited definition, as we now know that amyloids can also be functional. Therefore, the field had shifted to biophysical definitions, and mainly that amyloid fibrils display the cross- β fiber diffraction pattern when examined with X-rays.

The cross- α fibril was observed for the first time in PSM α 3 and was referred in the original paper¹⁵ and the current one as “amyloid-like”. We wish to stress the parallels to canonical amyloids. The cross- α PSM α 3 fibrils are cytotoxic, show unbranched morphology similar to that of canonical amyloids, bind ThT and display a characteristic amyloid-fibrillation curve. It also binds Congo red showing the spectral shift (we did not add this information to the original paper as Congo red is considered too promiscuous in binding to different fibrils). Most importantly, as amyloid definition is now mostly based on structural characteristics, we stress that the cross- α architecture is formed via mated sheets forming a tight and dry interface, wherein the “strands” (here α -helices) are oriented perpendicular to the fibril axis, just like in the cross- β structure. The surface complementarity is comparable between the cross- α and cross- β fibrils. The cross- α fibrils thus mostly resemble amyloids compared to any other biological fibrils. Furthermore, the result presented in the current paper, showing that PSM α 1 and PSM α 4, which are members of the same family and homologous to PSM α 3, form cross- β fibrils only strengthen the association of the cross- α PSM α 3 fibril to amyloids.

We do believe that PSM α 3 paved the way for extending the description of “amyloid-like” to include cross- α fibrils. We recently solved another cross- α structure from a eukaryotic source (yet unpublished result), and we are certain that more such fibrils will be revealed in the future.

Reviewer #3

“The article by Salinas et al. describes the amyloid assemblies formed by segments derived from the phenol-soluble modulin peptide family (PSMs). This work follows another article by the same group of authors on the structure of PSM-alpha3, a cytotoxic peptide. That structure of the 22-residue peptide revealed a tightly packed alpha helix with amyloid-like properties and was published in Science.

Through the use of x-ray micro diffraction, the group now reveals additional structures of potentially functional amyloids formed by PSM segments. The segments in question are overall shorter than that previously described for PSM-alpha3, and one that is a fragment

of that previously published structure. This article is of broad interest to the microbial amyloid community, revealing the variety of polymorphs potentially formed by bacterial amyloids.

The article raises technical and scientific questions, detailed below that could be addressed by minor revision."

Questions/comments:

1. *"The 'extreme' polymorphism referred to in the title appears only to apply to the segments associated with PSM-alpha3. In fact those of PSM-alpha1/4 are nearly identical in structure - by eye, the C-alpha RMSD appears near zero (not surprising given the high sequence identity between the two segments). This discordance plays two different stories to the reader, one in which two different PSMs look near identical, and the other where a single PSM is a shape-shifter."*

The extreme polymorphism relates to the PSM family in general, in which we observed cross- β , cross- α , β -rich hexameric structure and out-of-register β -sheets configurations in homologous sequences, and even within the same sequence. We attempted to better clarify this point in the revised manuscript (including new Supplementary Fig. 6). Our main conclusion is that different amyloid-like structures encode different functions. The near-identical structure of the PSM α 1 and PSM α 4 segments only reflects that both full-length polypeptides play a similar role in stabilizing the biofilm. PSM α 3 that forms cross- α fibrils is toxic to human cells, and the LFKFFK derivative, which we suggest to expand the array of activities by truncations, forms β -rich atypical and polymorphic fibrils that are antibacterial.

2. *"How do the fibril diffraction patterns shown in Figure 1 for PSM-alpha1/4 compare with simulated or measured fibril diffraction patterns from the PSM-alpha1/4 spine segments whose structures were determined here?"*

We added to the paper fibril diffraction measured at the ESRF synchrotron for the PSM α 1/ α 4 spine segments (revised Fig. 2c), showing the canonical cross- β signature, in accordance with the crystal structures.

3. *"The selection of the two spine segments from PSM-alpha1/4 seems ambiguous. Why not consider the region immediately following these spines (AIIDIF) which at least for PSM-alpha4 is largely hydrophobic?"*

The sequences selected were based on integrated information from several servers that predict amyloidogenic propensities (ZipperDB, Waltz, TANGO and Zyggregator). Also, since we already knew that the full length PSM α 1 and PSM α 4 form fibrils, we looked for a putative shared core that nucleates fibrillation and the most similar region was IIKVIK/IIKIIK. In PSM α 1 we also examined the sequences IIAGIIK and IIAGIIKVIK, which formed fibrils but did not form crystals.

Of note, amyloid-forming segments are not always hydrophobic (such as NNQQNY from yeast prion or VQIVIK from Tau). Eisenberg (and later our lab) observed interfaces of the mated sheets that contain polar residues forming multiple hydrogen bonds.

4. *“How does the structure of KLFKFFK compare to that of its shorter counterpart? Does the polymorphism seen for the shorter PSM-alpha3 segment depend on its length? If so, the evidence provided may not be representative of what is observed in longer, bioactive segments. The single micrograph of KLFKFFK provides little in the form of structural evidence to disambiguate these questions.”*

Reviewer #1 (comment #6) and Reviewer #2 (comment #1) raised similar questions and we correspondingly added experiments to examine the structural features of KLFKFFK. We could not have determined the crystal structure of KLFKFFK but observed that it forms polymorphic crystalline fibrils with some straight and some twisting morphologies (we now display both LFKFFK and KLFKFFK micrographs in Fig. 3). We added an analysis of the secondary structure of the fibrils using ATR-FTIR spectroscopy (lines 144-156), which showed similar spectra for LFKFFK and KLFKFFK with the presences of β -rich species (Supplementary Fig. 5). Specifically, the steric-zipper segment PSM α 1-IIKVIK shows a peak at 1621 cm^{-1} corresponding to rigid cross- β amyloid fibrils⁷⁻⁹, in accordance with the crystal structure (Fig. 2). Contrastingly, PSM α 3-LFKFFK shows two main peaks at 1622 cm^{-1} and 1633 cm^{-1} and PSM α 3-KLFKFFK shows a peak at 1633 cm^{-1} , with the latter indicating on more disordered fibers with absorbance which is typical of the bent β -sheets in proteins⁷⁻⁹, in accordance with the atypical and polymorphic β -rich crystal structures of LFKFFK. The similarity between LFKFFK and KLFKFFK fibrils, and their disordered and polymorphic nature compared to steric-zippers, further propose that the unique structural properties of these self-assembling peptides encode their antibacterial activity.

5. *“What does the comparison in Supplementary Figure 7 offer? Wouldn't we learn more from a comparison of the cross-alpha and cross-beta structures of PSM-alpha3?”*

PSM α 3 forms only the cross- α fibrils (and remains stable as α -helical in solution and in fibrils as observed by CD, FTIR and fibril diffraction). The LFKFFK derivative from

PSM α 3 forms the hexameric and out-of-register β -rich forms, but not cross- β . We believe that this derivative extends the arrays of functions of PSM α 3 rather than recapitulates its spine (as mentioned in our reply to Reviewer #2, comment #4). We revised the figure (now Supplementary Figure 6) in order to show the extensive polymorphism of PSM α s (cross- β , cross- α , β -rich hexameric structure and out-of-register β -sheets configurations).

6. *“The article touches on the labile nature of some of the segments investigated, but does not demonstrate any of the biochemical features associated with reversible amyloid behavior - phase separation, coalescence, etc. Do segments of PSM-alpha3 represent bonafide reversible amyloid? This area merits further investigation; its present description seems tangential and potentially premature.”*

We are actually not sure what it means to be “bonafide reversible amyloid” except from fibrils that can dissolve and reform upon changes in conditions. As far as we know, reversibility of amyloids was mostly discussed in the context of a pH- or temperature-induced reversibility of fibril formation, and was mostly related to regulation of activity, thus correlates with functional amyloids. Only some of those were directly correlated with phase separation or hydrogel formation. The vast majority of amyloid formation is not reversible as the free energy of a cross- β fibril is low compared to most other protein states. Specific and rare amyloid configurations can lead to reversibility. Some published examples of reversible amyloid are pH-induced switch showed for a P53 mutant¹⁶ and some peptide systems¹⁷, or androgen hormones that form fibrils that dissemble by the addition of a reducing agent¹⁸. Probably some other examples exist, but in many cases, a bonafide amyloid definition is missing. The most recent known examples of a reversible amyloid is the TAR DNA-binding protein 43 (TDP-43) that was found to aggregate both reversibly, to form stress granules, and irreversibly, yielding pathogenic amyloid¹⁹. Another recent example is the FUS functional amyloid involved in RNA granules (fibrillation associated with membraneless assemblies)^{4,5}. The full-length FUS forms hydrogels while reversibility was demonstrated by showing that fibrils of FUS segments are less thermostable compared to segments that form canonical steric-zippers. In the case of LFKFFK, we used a similar strategy to show reversibility via thermostability. Here the active species is the six-residue peptide (having antibacterial activity). In the electron micrographs, we did not observe any droplets or phase separation, but those are often difficult to characterize.

- 1 Eisenberg, D. S. & Sawaya, M. R. Structural Studies of Amyloid Proteins at the Molecular Level. *Annu. Rev. Biochem.* **86**, 69-95, doi:10.1146/annurev-biochem-061516-045104 (2017).
- 2 Lawrence, M. C. & Colman, P. M. Shape complementarity at protein/protein interfaces. *J. Mol. Biol.* **234**, 946-950, doi:10.1006/jmbi.1993.1648 (1993).
- 3 Sawaya, M. R. *et al.* Atomic structures of amyloid cross-beta spines reveal varied steric zippers. *Nature* **447**, 453-457, doi:10.1038/nature05695 (2007).
- 4 Luo, F. *et al.* Atomic structures of FUS LC domain segments reveal bases for reversible amyloid fibril formation. *Nat. Struct. Mol. Biol.*, doi:10.1038/s41594-018-0050-8 (2018).
- 5 Hughes, M. P. *et al.* Atomic structures of low-complexity protein segments reveal kinked beta sheets that assemble networks. *Science* **359**, 698-701, doi:10.1126/science.aan6398 (2018).
- 6 Balbirnie, M., Grothe, R. & Eisenberg, D. S. An amyloid-forming peptide from the yeast prion Sup35 reveals a dehydrated [bgr]-sheet structure for amyloid. *Proc. Natl Acad. Sci. USA* **98**, 2375-2380 (2001).
- 7 Sarroukh, R., Goormaghtigh, E., Ruyschaert, J. M. & Raussens, V. ATR-FTIR: a "rejuvenated" tool to investigate amyloid proteins. *Biochim. Biophys. Acta* **1828**, 2328-2338, doi:10.1016/j.bbame.2013.04.012 (2013).
- 8 Zandomenighi, G., Krebs, M. R., McCammon, M. G. & Fandrich, M. FTIR reveals structural differences between native beta-sheet proteins and amyloid fibrils. *Protein Sci.* **13**, 3314-3321 (2004).
- 9 Moran, S. D. & Zanni, M. T. How to Get Insight into Amyloid Structure and Formation from Infrared Spectroscopy. *J. Phys. Chem. Lett.* **5**, 1984-1993, doi:10.1021/jz500794d (2014).
- 10 Jang, K. S., Park, M., Lee, J. Y. & Kim, J. S. Mass spectrometric identification of phenol-soluble modulins in the ATCC(R) 43300 standard strain of methicillin-resistant *Staphylococcus aureus* harboring two distinct phenotypes. *Eur. J. Clin. Microbiol. Infect. Dis.* **36**, 1151-1157, doi:10.1007/s10096-017-2902-2 (2017).
- 11 Gonzalez, D. J. *et al.* Novel phenol-soluble modulin derivatives in community-associated methicillin-resistant *Staphylococcus aureus* identified through imaging mass spectrometry. *J. Biol. Chem.* **287**, 13889-13898, doi:10.1074/jbc.M112.349860 (2012).
- 12 Gonzalez, D. J. *et al.* Phenol soluble modulin (PSM) variants of community-associated methicillin-resistant *Staphylococcus aureus* (MRSA) captured using mass spectrometry-based molecular networking. *Mol. Cell Proteomics* **13**, 1262-1272, doi:10.1074/mcp.M113.031336 (2014).
- 13 Deplanche, M. *et al.* Phenol-soluble modulin alpha induces G2/M phase transition delay in eukaryotic HeLa cells. *FASEB J.* **29**, 1950-1959, doi:10.1096/fj.14-260513 (2015).
- 14 Joo, H. S., Cheung, G. Y. & Otto, M. Antimicrobial activity of community-associated methicillin-resistant *Staphylococcus aureus* is caused by phenol-soluble modulin derivatives. *J. Biol. Chem.* **286**, 8933-8940, doi:10.1074/jbc.M111.221382 (2011).
- 15 Tayeb-Fligelman, E. *et al.* The cytotoxic *Staphylococcus aureus* PSMalpha3 reveals a cross-alpha amyloid-like fibril. *Science* **355**, 831-833, doi:10.1126/science.aaf4901 (2017).
- 16 Lee, A. S. *et al.* Reversible amyloid formation by the p53 tetramerization domain and a cancer-associated mutant. *J. Mol. Biol.* **327**, 699-709 (2003).
- 17 Yamaguchi, K., Kamatari, Y. O., Fukuoka, M., Miyaji, R. & Kuwata, K. Nearly reversible conformational change of amyloid fibrils as revealed by pH-jump experiments. *Biochemistry* **52**, 6797-6806, doi:10.1021/bi400698u (2013).
- 18 Asencio-Hernandez, J. *et al.* Reversible amyloid fiber formation in the N terminus of androgen receptor. *Chembiochem : a European journal of chemical biology* **15**, 2370-2373, doi:10.1002/cbic.201402420 (2014).
- 19 Guenther, E. L. *et al.* Atomic structures of TDP-43 LCD segments and insights into reversible or pathogenic aggregation. *Nat. Struct. Mol. Biol.*, doi:10.1038/s41594-018-0064-2 (2018).

REVIEWERS' COMMENTS:

Reviewer #2 (Remarks to the Author):

My original review of this paper asked for certain clarifications and further discussions, although I strongly felt that the novel structures reported here were of interest. I am now very happy with the careful responses to my questions and comments and the improvements made to the manuscript.

Reviewer #3 (Remarks to the Author):

The authors are correct in that the exact definitions of some terms in the amyloid field are not yet fully established. Given their role at the front lines of this field and their power to illuminate the atomic basis for amyloid phenomena, it is incumbent upon them to rigorously prove assertions that stretch our current definitions.

The authors appear to be taking steps toward this goal and have adequately revised their manuscript for publication.